# Prediction of Machine Inactivation Status Using Statistical Feature Extraction and Machine Learning

**Taing Borith, Sadirbaev Bakhit, Aziz Nasridinov and Kwan-Hee Yoo \***

Department of Computer Science, Chungbuk National University, Cheongju-Si 28644, Korea;
tborith@gmail.com (T.B.); sadirbaev@cbnu.ac.kr (S.B.); aziz@chungbuk.ac.kr (A.N.)
**\*** Correspondence: khyoo@cbnu.ac.kr

**Abstract:** In modern manufacturing, the detection and prediction of machine anomalies, i.e., the inactive state of the machine during operation, is an important issue. Accurate inactive state detection models for factory machines can result in increased productivity. Moreover, they can guide engineers in implementing appropriate maintenance actions, which can prevent catastrophic failures and minimize economic losses. In this paper, we present a novel two-step data-driven method for the non-active detection of industry machines. First, we propose a feature extraction approach that aims to better distinguish the pattern of the active state and non-active state of the machine by multiple statistical analyses, such as reliability, time-domain, and frequency-domain analyses. Next, we construct a method to detect the active and non-active status of an industrial machine by applying various machine learning methods. The performance evaluation with a real-world dataset from the automobile part manufacturer demonstrates the proposed method achieves high accuracy.

**Keywords:** statistical feature extraction; machine learning; machine non-active state

## 1. Introduction

In modern manufacturing, machines in factories operate continuously for 24 h a day to fulfill the production requirements. In such cases, continuous production or continuous flow process enables factories to manufacture, produce, or process materials without interruptions [1]. However, massive industrial processes can be detrimental to machines. Various unstable operations, such as abnormal events, failures, or non-active states, can occur in the machines. These non-active states can significantly affect continuous production in the factories, incurring a significant loss to the fabricator and decreasing production rates. Besides, constant interruptions in manufacturing make it difficult for fabricators to fulfill their commitments to the consumers [2]. Therefore, predicting non-active states of machines in advance is essential to reduce machine faults and increase productivity.

As industrial processes become more and more complex, there is a need to automate the detection and prediction of unstable operations of machines as maintenance decisions need to be taken quickly to avoid costly interruptions [3]. However, the automation of such tasks was often challenging in the past due to the lack of sensors and data-driven technologies. On the other hand, with the rapid development of data acquisition devices based on sensor technologies, factories now collect a massive amount of data related to the state of the machines. We can apply various data-driven models to analyze the sensor data and obtain meaningful insights that could be helpful in achieving automation in detecting and predicting the unstable operations of machines [4,5]. Thus, data-driven maintenance of machines has become the mainstream solution for solving various manufacturing issues.

Most studies on data-driven maintenance of machines in the literature are based on statistical, machine learning, and deep learning methods. Well-known statistical analysis methods, such as Bayesian algorithms [6] and hidden Markov models [3], have been proved to be an effective solution for

estimating time-to-failure (TTF) and detection faults in machines. On the other hand, machine learning methods, such as k-nearest neighbors (kNN) [7] and Artificial Neural Networks (ANN) [8], have been actively utilized in predicting unstable operations of machines, especially in the presence of a large amount of data. Deep learning methods, such as a dual-path recurrent neural network (RNN) [9] and Back Propagation Neural Network (BPNN) [10], have recently shown strong performances in detecting the early mechanical fault in various manufacturing processes.

Most of the methods mentioned above focus on detecting or predicting large-scale failures in machine operation that can be potentially harmful but not frequent. On the other hand, there could be various non-active states associated with machines that occur more frequently and can be equally damaging. For example, machines may become non-active due to the supply of poor materials, wear, disconnection, burnout, or equipment related malfunctioning in sensors, motors, and control switches. Predicting the non-active status of machines can bring to the benefits efficient production of manufacturing machines, such as reduction of long-term machine failures and maintenance cost, and increasing machine lifetime and production. Thus, in this paper, we focus on predicting non-active states of the machines using statistical feature extraction and machine learning. More specifically, we make the following contributions in the paper:

- We propose to extract various statistical features from the raw data. We first extract so-called reliability features using Weibull, lognormal, and exponential distributions. These distributions are mainly used for life data analysis to estimate the lifetime of a machine. Further, we propose a method to compute a new feature, i.e., the machine status tracking value (MSTV), which can distinguish the active and non-active patterns of a machine more effectively. Lastly, to further improve the accuracy of predicting non-active states of the machines, we utilize MSTV to extract time and frequency domain features.
- After statistical features have been extracted, learning models for predicting non-active states of the machines are constructed by combining the raw data from the sensor device and extracted features. Specifically, we construct predictive models using state-of-the-art machine learning methods, namely decision tree, kNN, random forests, and linear support vector machines (SVM).
- We evaluated the performance of the proposed method through extensive experiments. In experiments, we measured the accuracy and error rate of state-of-the-art machine learning methods in classifying the active and non-active states of the machine with 17 features and 86,400 observations. The real-world dataset was obtained by automobile part manufacturer. The experiment results demonstrate that linear SVM achieves a 98% accuracy compared with the other models, which can be considered as a promising result in performing predictive maintenance of the machines.

The rest of the paper proceeds as follows. Section 2 discusses studies related to the detection and prediction of unstable operations in machines. Section 3 describes the proposed method. Section 4 presents the performance evaluation. Section 5 concludes the paper and discusses future work.

## 2. Related Studies

In this section, we discuss the related studies that focus on the detection and prediction of unstable operations in machines. We can classify these studies into three categories: (1) Studies based on statistical analysis, (2) studies based on machine learning, and (3) studies based on deep learning.

There have been several studies that used statistical analysis for the detection and prediction of unstable operations in machines. For example, Wu et al. [6] proposed a three-step degradation TTF prognostic method for rolling element bearings (REBs) in an electrical machine. The three-step degradation approach includes degradation feature extraction, degradation feature reduction, and TTF prediction. To detect degradations during degradation feature extraction, multiple degradation features, including statistical, intrinsic energy, and fault frequency features, were extracted. In the degradation feature reduction step, the authors performed a feature fusion using dynamic principle

component analysis (DPCA) and Mahalanobis distance. In the final step, the authors utilized the exponential regression algorithm to compute the local degradation model and the empirical Bayesian algorithm to calculate the global TTF prediction. The experimental results demonstrated that the proposed method achieves a good performance of the TTF predictions compared with the existing methods. Boutros and Liang [3] proposed a method for detecting and diagnosing mechanical faults in machining processes and rotating machinery using hidden Markov models. The experiment results with cutting tool and bearing monitoring cases demonstrate that the fault severity classification was greater than 95%. The authors also analyzed fault localization using a new concept called location index. The results of the analysis demonstrated that the proposed method classifies the various fault location in bearing monitoring with an accuracy of 96%.

Machine learning techniques are efficient in analyzing a large amount of data and derive various useful patterns. Considering that factories collect more and more data related to the machine state, we can efficiently utilize machine learning techniques in detecting and predicting unstable operations of machines. A similar study was carried out by Zhou et al. [7]. In this study, the authors introduced an approach for fault detection using random projections and the RP-kNN rule for semiconductor manufacturing processes. The authors utilized random projection, which is a type of dimension reduction method, to maintain the pairwise distances between samples in a random subspace and established a detection model with the kNNs algorithm. Besides, the authors illustrated PC-kNN [11] methods in a principal component subspace (PCS) to prove that the RP-kNN was more effective at preserving pairwise distances in the PCS. In the experiment, it was discovered that the detection model constructed using RP-kNN performed better, with approximately 82% accuracy compared with 77% of the PC-kNN model. Mazhar et al. [8] proposed to estimate the remaining life of used components in consumer products with Weibull distribution and ANN. The proposed model was constructed in two stages. In the first stage, the authors used Weibull distribution to analyze the behavior of components for reuse. Specifically, the Weibull distribution was applied to the TTF data to assess the mean life of the component. In the second phase, the authors developed an ANN model, a multilayer feedforward backpropagation neural network, to analyze the degradation and condition monitoring data. The performance evaluation with life cycle data from a washing machine demonstrates that the proposed model achieved an accuracy of approximately 86.6%.

With the recent advances in deep learning, many researchers have proposed various data-driven methods to detect abnormal events in a machine or provide fault prognostics for factory machines. For example, Shenfield et al. [9] presented an intelligent real-time fault detection method to provide early detection of developing problems under variable operating conditions. The authors proposed a novel dual-path RNN-WDCNN to diagnose rolling element bearing faults in manufacturing. RNN-WDCNN combines elements of RNNs [12] and convolutional neural networks (CNN) [13] to capture distant dependencies in time series data and suppress high-frequency noise in the input signals. The authors mentioned that RNN-WDCNN outperformed current state-of-art methods in both domain adaptation and noise rejection tasks. Luo et al. [10] proposed a novel method for early fault detection under time-varying conditions using a deep learning model called BPNN. A deep learning model was constructed to automatically select the impulse responses from the vibration signals in the long-term running of 288 days. Dynamic properties were then identified from the selected impulse responses to detect the early mechanical fault under time-varying conditions. The authors showed that the experimental results proved that the method was not affected by time-varying conditions and showed considerable potential for early fault detection in manufacturing. Iqbal et al. [14] presented a novel approach for automated Fault Detection and Isolation (FDI) based on deep learning. Consequently, the approach predicted the future states of a system based on its previous behavior while considering significant noise in the data. The approach can automatically learn complex real-world patterns to identify abnormal conditions. The proposed method was shown to outperform other established FDI methods. The authors claimed that the approach can successfully diagnose and locate multiple classes of faults under real-time working conditions.

## 3. The Proposed Method

### 3.1. Data Preparation

We first illustrated the raw data used to detect the non-active state of a machine. Our dataset contained real-world information obtained from an automobile part manufacturing located in South Korea. This factory applied the continuous flow process in which one-line batches were generated using various machines to produce automobile parts. All the machines operated for 24 h a day. Here, the machines were equipped with programmable logic controller (PLC) sensors that produced the data related to the machine state. Table 1 presents a detailed description of the raw dataset.

**Table 1.** Detailed information of raw data gathered from PLC.

| Type | Description | Data Type |
|---|---|---|
| Machine name | Indicate machine name | String |
| Date and time | Indicate date and time of all events in the machine | Time |
| Machine status | Indicate machine states: Run, Wait, Stop, Manual, Offline | Int |
| Machine state duration | Duration of machine state from one state to another | Int |
| Alarm status | Alarm status: 1 = alarm triggered; 0 = alarm not triggered | Binary |
| Alarm duration | Duration of alarm | Int |
| Continuous good product | Cumulative number of good products produced | Int |
| Continuous NG product | Cumulative number of poor-quality products produced | Int |

From the raw data provided in Table 1, we can see that a machine was managed as one of the following five states: Run, wait, stop, offline, and manual. The run state indicated that the current machine was operating appropriately. The wait state showed that the current machine was waiting for the previous machine to finish its job. These 2 states were defined as active. The stop state indicated that the current machine was stopped, and the manual state indicated that the current machine was operating manually. Finally, the offline state indicated that the machine was not connected to any network such that data gathering from the machine was not achieved. These 5 states of a machine were referred to as a machine state. Among these 5 states, we can consider run and wait as active states. On the other hand, stop, offline, and manual were considered as non-active states. Figure 1 shows the machine operating processes for a period of the day in the automobile manufacturer displayed through the Web monitoring system. In the figure, green and yellow colors indicated the active state of the machine, and red indicated non-active states of the machine.

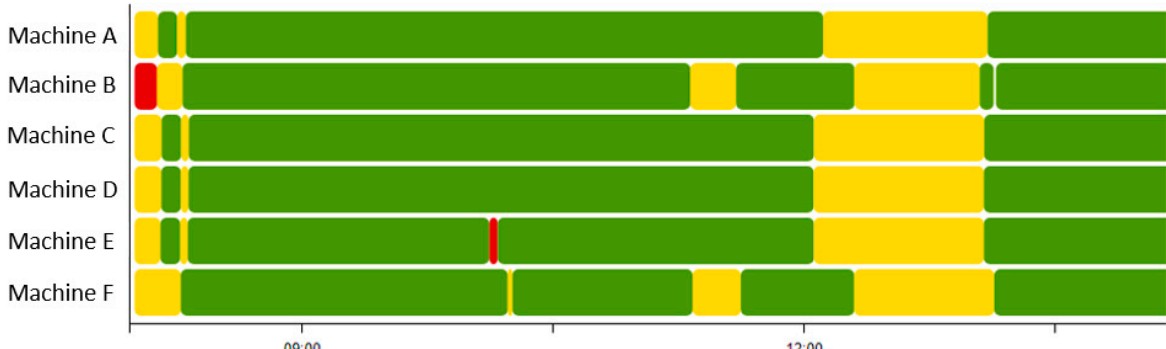

**Figure 1.** Examples of machine operating processes.

From Table 1, we can also observe that we have another type of data related to alarms that were triggered when a machine entered into a non-active state. Specifically, the alarm signal notified the machine operators of an abnormal situation. Besides, considering that a machine was operated to produce certain products, we could monitor the machine state by the number of produced good and NG products.

*3.2. Features Extraction Using Statistical Analysis*

In this paper, we proposed to extract reliability features, MSTV features and time and frequency domain features from the raw data described in Section 3.1. In the subsequent subsections, we will describe each feature in detail.

3.2.1. Reliability Features

In this study, the reliability measures that we used for extracting features from raw data included Weibull, lognormal, and exponential distributions. Reliability analysis was a statistical measure for life data [15,16]. It was determined by deriving the proportion of methodical variation in a scale, which can be performed by determining the association between the scores obtained from divergent administrations of the scale. Therefore, if the association in reliability analysis was high, the scale yielded consistent results and was, therefore, reliable. To compute the reliability of the machines in a manufacturing line, 2 measures, called the time between non-active states (TBNA) and mean time between non-active (MTBNA) states of the machines, were calculated. Figure 2 illustrates the structure of the reliability analysis.

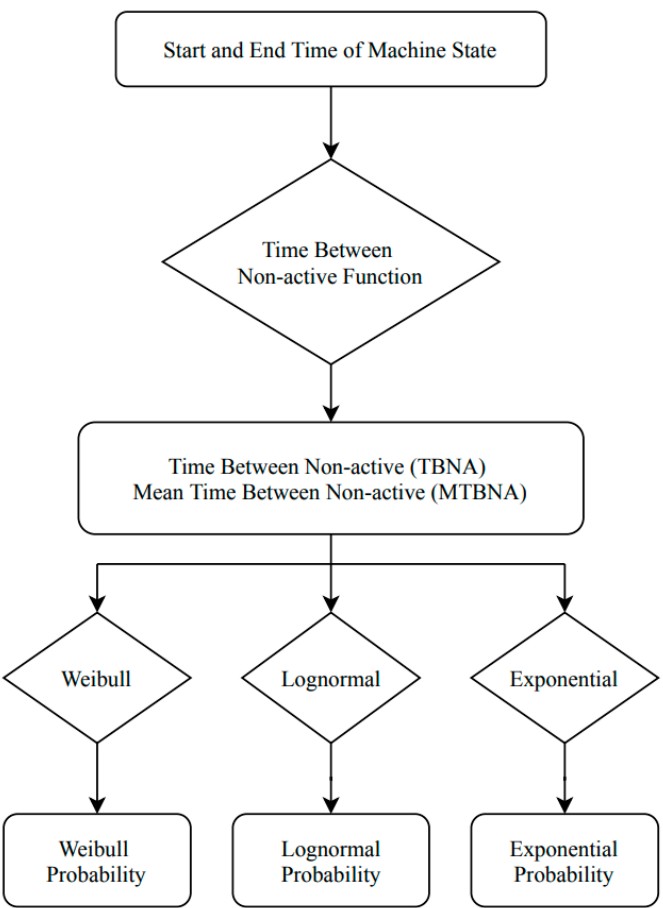

**Figure 2.** Structure of feature extraction in reliability analysis.

Based on the TTF [17,18] mathematical formulation, we can define the *TBNA* state, which is the elapsed time between the non-active states of the machines, and *MTBNA* as follows.

$$TBNA = start\ of\ non-active\ time - start\ of\ active\ time \tag{1}$$

$$MTBNA = \frac{\sum TBNA}{number\ of\ non-active\ states} \tag{2}$$

Once *TBNA* and *MTBNA* are calculated, we extract features from raw data using Weibull, lognormal, and exponential distributions. We first obtained the Weibull distribution, which was one of the most widely used lifetime data analyses for reliability engineering [19]. It can flexibly model various types of lifetime distributions [20]. In this paper, we used a 2-parameter Weibull distribution, which had scale and shape parameters. Here, the scale parameter was denoted as $\eta$, and the shape parameter was denoted as $\beta$ [21]. When $\beta$ was less than 1, the distribution showed a decreasing failure rate over time [22]. When $\beta$ was 1, the distribution had a constant failure rate. When the $\beta$ parameter was greater than 1, the failure rate increased over time [21]. We estimated the appropriate scale and shape parameters of the Weibull distribution by using the TBNA with maximum likelihood estimation (MLE) method. More specifically, the MLE formula [23] for determining the parameters of the Weibull distribution is written as follows:

$$\beta = \left[ \frac{\sum_{i=1}^{n} t_i \ln(t_i)}{\sum_{i=1}^{n} t_i} - \frac{\sum_{i=1}^{n} \ln(t_i)}{n} \right]^{-1}$$
$$\eta = \left( \frac{1}{n} \sum_{i=1}^{n} t_i \right)^{\frac{1}{\beta}} \tag{3}$$

In Equation (3), $t_i$ was TBNA calculated in Equation (1), and $i$ and $n$ are the numbers of non-zero data points. Once the shape parameter $\beta$ and scale parameter $\eta$ of the Weibull distribution was calculated using Equation (3), we could estimate the non-active state rate of the machine using the probability density function of the Weibull as follows:

$$f(t) = \frac{\beta}{\eta} \left( \frac{t}{\eta} \right)^{\beta-1} e^{-(t/\eta)^{\beta}} \tag{4}$$

Figure 3 illustrates the fitting curve result of the Weibull distribution with 2 parameters set as $\beta = 0.71$, eta $\eta = 144.02$, and the mean life of the non-active state was set as 179.6, which means that the non-active state of the machine will occur when the machine was operating for approximately 179 min. In Figure 3, the *x*-axis indicated the elapsed time (in minutes) from which the machine was active, and the *y*-axis indicated the probability of the non-active state of the machine. From the figure, we can observe the relationship between active and non-active states of the machine. For example, when the probability of non-activation of a facility was 0.8, it means that about 280 min have passed from the time it was activated.

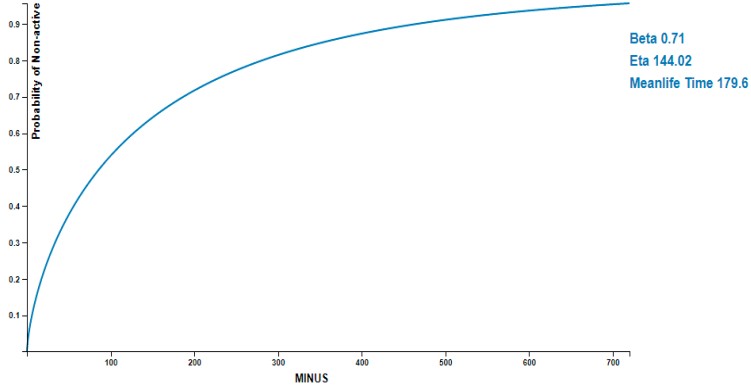

**Figure 3.** Result of the fitting curve of Weibull distribution.

Further, we obtained the lognormal distribution, which was a constant probability distribution of random variables [24]. This distribution was also widely used to model the lives of units whose failure modes were of the fatigue–stress nature [25]. Additionally, the lognormal distribution complements

the Weibull distribution well for modeling the reliability of a machine. The formula of the lognormal distribution is written as follows:

$$f(t') = \frac{1}{\sigma' \sqrt{2\pi}} e^{-\frac{1}{2}\left(\frac{t'-\mu'}{\sigma'}\right)^2}, \tag{5}$$

In Equation (5), $t'$ is $ln(t)$, where $t$ is TBNA calculated in Equation (1), $\mu'$ the logarithm of MTBN calculated in Equation (2), and $\sigma'$ the standard deviation of the natural logarithms of TBNA.

Figure 4 shows the result of lognormal distribution probability. In Figure 4, the $x$-axis indicated the elapsed time (in minutes) from which the machine was active, and the $y$-axis indicated the probability of the non-active state of the machine obtained using the lognormal distribution in Equation (5). From the figure, we can observe that a machine enters active and non-active states over time. Here, if the value on the $y$-axis was 1, the machine was activated, and if the value was close to 0.0, the probability of non-activation increased. In other words, we can see that lognormal distribution is used to describe the probability of a specific event occurring.

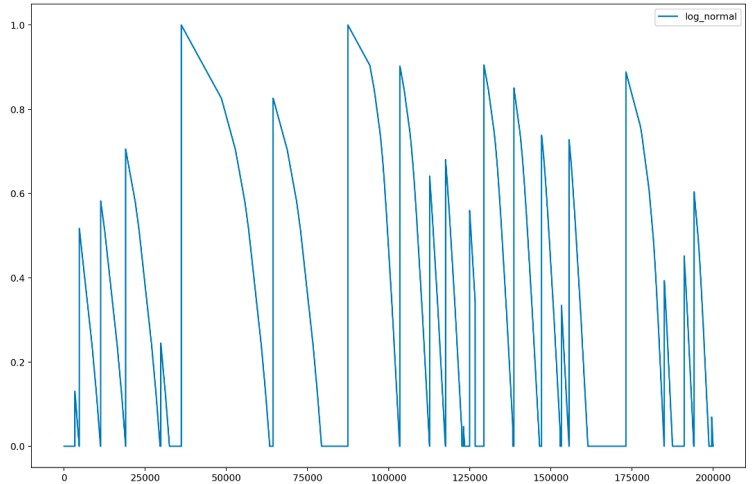

**Figure 4.** Result of lognormal distribution.

Similar to the Weibull and lognormal distributions, the exponential distribution was typically used in reliability engineering and exhibited a simple distribution [26]. The exponential distribution was utilized to model the behavior of units that have a constant failure rate. The primary probability density function for the exponential distribution is expressed as follows.

$$f(t) = \lambda e^{-\lambda t} = \frac{1}{m} e^{-\frac{1}{m}t}, \ t \geq 0, \ \lambda > 0, \ m > 0, \tag{6}$$

In Equation (6), $\lambda = 1/m$ is the constant rate in the non-active state per unit of measurement, $m$ the mean TBNA, and $t$ the TBNA during the machine operation.

Figure 5 illustrates the result of the exponential distribution. In Figure 5, the $x$-axis indicated the elapsed time (in minutes) from which the machine was active, and the $y$-axis indicated the probability of a non-active state of the machine obtained using exponential distribution in Equation (6). Similar to Figure 4, Figure 5 demonstrated that a machine entered active and non-active states over time. Unlike lognormal distributions, the exponential distribution model the time elapsed between events and represented the amount of time until the machine enters into a non-active state.

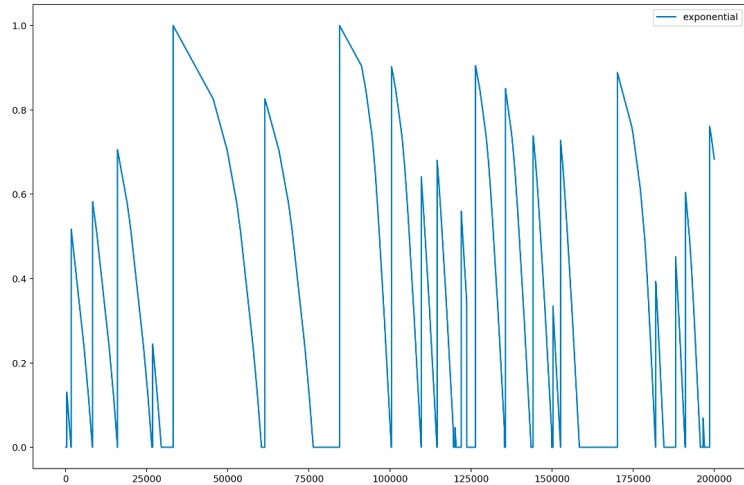

**Figure 5.** Result of exponential distribution results.

In feature extraction using reliability analysis, we extracted 4 features, including TBNA, Weibull probability, lognormal probability, and exponential probability. These 4 features will be used later to model the prediction of the non-active state of machines. Table 2 shows the 4 elements extracted from the reliability analysis.

**Table 2.** Reliability Feature Extraction.

| Feature | Description | Data Type |
|---|---|---|
| Time Between Non-active States | Time elapsed between non-active states | Int |
| Weibull Distribution | | Float |
| Lognormal Distribution | Probability of non-active states occurring during machine operation | Float |
| Exponential Distribution | | Float |

### 3.2.2. MSTV Features

To distinguish the active and non-active patterns of the machine more effectively, we proposed a method to combine various essential factors from the raw dataset related, i.e., the alarm duration (AD), machine state duration (MSD), continuous good product (CGP), and continuous NG product (CNP) to extract new features that can learn the machine's behavior during operation. We call these features an MSTV. Figure 6 shows the architecture of the proposed feature extraction.

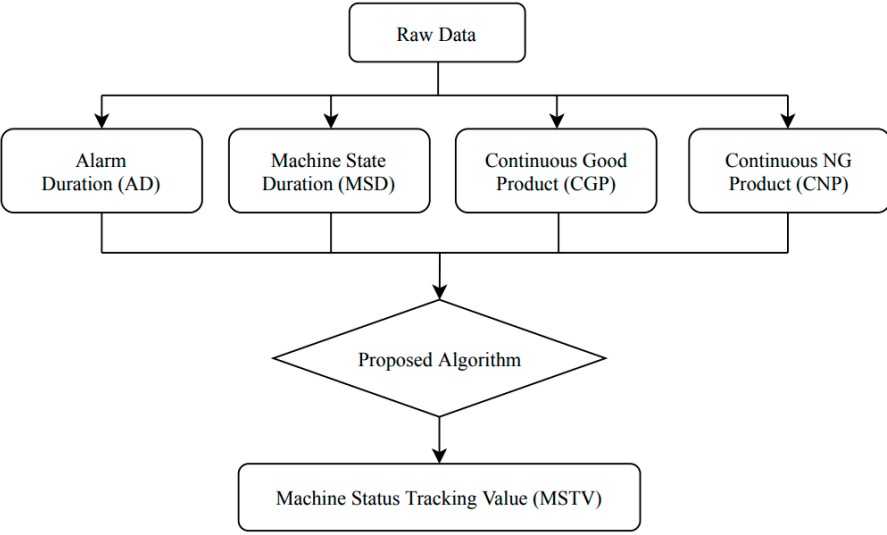

**Figure 6.** Architecture of proposed feature extraction.

Kang et al. [27] proposed the detection of significant arms using outlier detection algorithms, which indicated the importance of alarm for tracking the behavior of a machine, particularly during the occurrence of an unusual event. Hence, in this experiment, we retrieved the AD during operation as a combined feature for extracting the new useful feature. Meanwhile, the machine duration (MD) provided beneficial information for verifying the behavior of the machine because it indicated the time required by the machine to change from one state to another. Recall from Section 3.1 that a machine can have the following 5 states: Run, wait, stop, offline, and manual, among which run and wait were considered as active states, and stop, offline and manual were considered as non-active states. The CGP means the number of high-quality products produced. The CNP was the quantity of poor-quality products produced.

Based on these 4 features, we define *MSTV* as follows:

$$MSTV = \Sigma(AD \times w1 + MSD \times w2 + CGP \times w3 + CNP \times w4) \times \frac{1}{n}, \tag{7}$$

In Equation (7), $w1$, $w2$, $w3$, and $w4$ are weight values assigned to each factor, respectively. Here, the weight values indicate the importance of the factors that influence the non-active state of the machines. In this case, the importance was determined through discussions with the operators in the factory, and the most important factors were determined as the MSD, followed by the AD. Next was the number of normal products continuously and the number of abnormal products continuously produced by the machine. In consideration of these factors, in this paper, weights were given as 2, 3, 1.5, and 1, respectively.

Figure 7 shows the results of MSTV feature extraction. In Figure 7, the *x*-axis indicated the elapsed time (in minutes) from which the machine was active, and the *y*-axis indicated the probability of a non-active state of the machine obtained using MSTV in Equation (7). From the figure, we can observe that if the value was close to 0 (indicated in red rectangles), the probability of non-activation increased. On the other hand, we can see that the active state (green rectangle) occurred as the probability curve of MSTV goes higher. As shown in Figure 7, after employing the proposed feature extraction approach to extract the MSTV, the patterns of the active and non-active states of the machine can be distinguished well by combining 4 indispensable elements from the raw dataset to extract new utilitarian features.

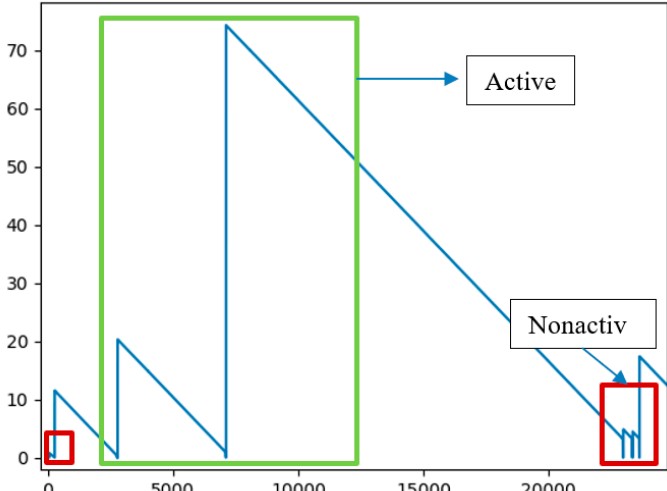

**Figure 7.** Results of machine status tracking value (MSTV) feature extraction.

### 3.2.3. Time and Frequency Domain Features

Once MSTV features were obtained using Equation (7), it was applied to the time and frequency domain analyses to extract the other 8 features to enhance the accuracy of our non-active state prediction model. The time-domain analysis employed mathematical functions to analyze physical signals or



time-series data with respect to time [28]. Additionally, the time-domain represented the change of a signal with time. Frequency analysis was the statistical method used for measuring the signals with respect to frequency. Frequency analysis typically involved the central tendency, dispersion, and percentiles [29,30]. Figure 8 shows the structure of feature extraction in the time-frequency and domain frequency features.

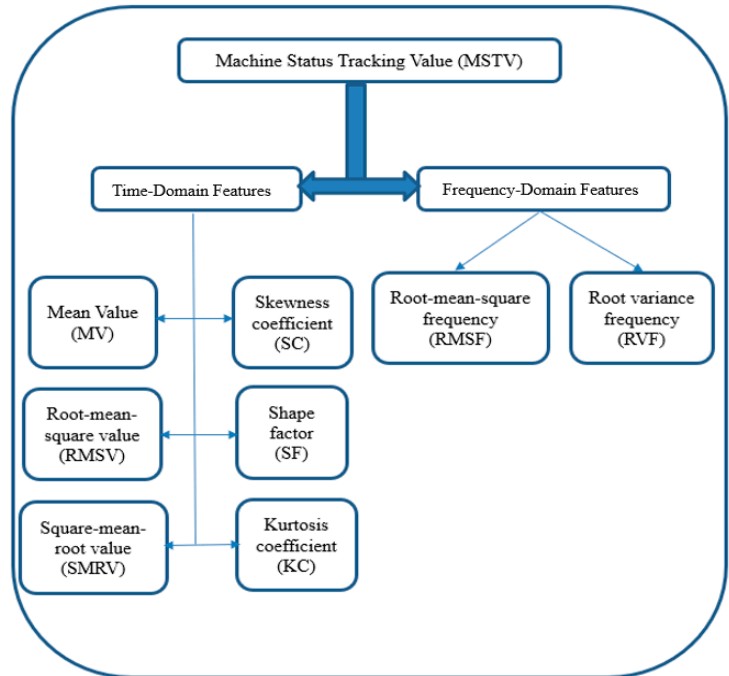

**Figure 8.** Structure of feature extraction in Time-domain and Frequency-domain features.

In the time-domain analysis, several statistical features were used that encompassed a wide range of popular time-domain analyses [6]. These statistical features included the mean value (MV), root mean square value (RMSV), square mean root value (SMRV), kurtosis coefficient (KC), shape factor (SF), and skewness coefficient (SC), which will be utilized to extract the new features from the MSTV. In addition, the root mean square frequency (RMSF) and root variance frequency (RVF), which are obtained from the frequency-domain analysis, will be used to extract two other features from the MSTV data. The details of 8 statistical methods for extracting features from the MSTV are listed in Table 3.

**Table 3.** Eight Statistical Features of MSTV.

| Features | Equation | Data Type |
|:---:|:---:|:---:|
| | Time-Domain Analysis | |
| Mean Value | $X_{mv} = \frac{1}{N} \sum\limits_{i=1}^{N} X(t_i)$ | Float |
| Root-mean-square Value | $X_{rmsv} = \sqrt{\frac{1}{N} \sum\limits_{i=1}^{N} X^2(t_i)}$ | Float |
| Square-mean-root Value | $X_{smrv} = \left[ \frac{1}{N} \sum\limits_{i=1}^{N} \sqrt{|X(t_i)|} \right]^2$ | Float |
| Skewness Coefficient | $X_{sc} = \frac{1}{X_{rmsv}{}^3} \sum\limits_{i=1}^{N} (X(t_i) - X_{mv})^3$ | Float |
| Kurtosis Coefficient | $X_{kc} = \frac{1}{X_{rmsv}{}^4} \sum\limits_{i=1}^{N} (X(t_i) - X_{mv})^4$ | Float |
| Shape Factor | $X_{sf} = \frac{X_{rmsv}}{\frac{1}{N} \sum_{i=1}^{N} |X(t_i)|}$ | Float |

**Table 3.** *Cont.*

| Features | Equation | Data Type |
|---|---|---|
| | Frequency-Domain Analysis | |
| Root Mean Square Frequency | $X_{rmsf} = \sqrt{\dfrac{\sum_{i=2}^{N} X^2(t_i)}{4\pi^2 \sum_{i=1}^{N} X^2(t_i)}}$ | Float |
| Root Variance Frequency | $X_{rvf} = \sqrt{\dfrac{\sum_{i=2}^{N} X^2(t_i)}{4\pi^2 \sum_{i=1}^{N} X^2(t_i)} - \left(\dfrac{\sum_{i=1}^{N} X^2(t_i)}{2\pi \sum_{i=1}^{N} X^2(t_i)}\right)^2}$ | Float |

### 3.3. Data Normalization

Data normalization was one of the essential preprocessing steps for many machine learning algorithms. Raw data and extracted features often have different scales. In other words, the variation between features can be inconsistent, which may result in many problems during data training, including lowering accuracy and increasing the training time. The normalization process for data contributed positively to preparing data that were suitable for training [31]. Moreover, data normalization can scale the data in the same range of values for each input feature to minimize bias in the input data. Furthermore, data normalization can also accelerate the training time by starting the training operation for each feature within the same scale. Various types of data normalization methods can be employed to reduce the bias of the training data, such as the Min-Max Normalization, Z-Score Normalization, Sigmoid Normalization, and others.

In our case, after feature extraction using statistical analysis described in Section 3.2, 17 input features were obtained, including raw data from the sensor device. The variations between the input features were large and diverse, which may cause problems in the training phase. Thus, we needed to normalize the data. We used Min-Max Normalization that rescaled the features from one range of values to a new range of values. Typically, the features were rescaled within a range of 0 to 1 or from $-1$ to 1. Table 4 shows the application Min-Max Normalization on a sample training dataset, which is calculated using the following formula:

$$x' = (x_{max} - x_{min}) \times \frac{(x_i - x_{min})}{x_{max} - x_{min}} + x_{min} \tag{8}$$

In Equation (8), *i* is *i*th element of the dataset.

**Table 4.** Sample training dataset.

| No | MSD | CGP | CNP | TBNA | Weibull Probability | Exponential Probability | Lognormal Probability | MSTV | MV | RMSV | SMRV | SC | KC | SF | RMSF | RVF |
|---|---|---|---|---|---|---|---|---|---|---|---|---|---|---|---|---|
| 1 | 0.396938 | 0.603901 | 0.073171 | 0.396938 | 0.711354 | 0.603363 | 0.999969 | 0.437343 | 0.437343 | $1.91286 \times 10^{-1}$ | 0.437343 | $8.321303 \times 10^{-15}$ | $1.686249 \times 10^{-19}$ | $8.364994 \times 10^{-2}$ | 0.046755 | $9.948226 \times 10^{-3}$ |
| 2 | 0.000000 | 0.000000 | 0.000000 | 0.000000 | 0.000000 | 0.000000 | 0.000000 | 0.000000 | 0.000000 | 0.000000 | 0.000000 | 0.000000 | 0.000000 | 0.000000 | 0.000000 | 0.000000 |
| 3 | 0.234286 | 0.342357 | 0.024390 | 0.234322 | 0.451382 | 0.332008 | 0.999919 | 0.256797 | 0.256797 | $6.594449 \times 10^{-2}$ | 0.256797 | $4.110406 \times 10^{-14}$ | $1.418555 \times 10^{-18}$ | $1.693450 \times 10^{-2}$ | 0.034228 | $2.510931 \times 10^{-3}$ |
| 4 | 0.000110 | 0.000000 | 0.000000 | 0.000000 | 0.000000 | 0.000000 | 0.000000 | 0.000080 | 0.000080 | $6.361941 \times 10^{-9}$ | 0.000080 | $1.371742 \times 10^{-3}$ | $1.524158 \times 10^{-4}$ | $5.074397 \times 10^{-13}$ | 0.000014 | $4.571769 \times 10^{-11}$ |
| 5 | 0.460787 | 0.421576 | 0.032520 | 0.460787 | 0.881584 | 0.823336 | 0.999987 | 0.482576 | 0.482576 | $2.328801 \times 10^{-1}$ | 0.482576 | $6.193819 \times 10^{-15}$ | $1.137482 \times 10^{-19}$ | $1.123824 \times 10^{-1}$ | 0.215370 | $5.579461 \times 10^{-2}$ |
| 6 | 0.026436 | 0.000000 | 0.000000 | 0.000000 | 0.000000 | 0.000000 | 0.000000 | 0.012762 | 0.012762 | $1.628657 \times 10^{-4}$ | 0.012762 | $3.348980 \times 10^{-10}$ | $2.325680 \times 10^{-13}$ | $2.078473 \times 10^{-6}$ | 0.000573 | $1.038138 \times 10^{-7}$ |
| 7 | 0.001248 | 0.251990 | 0.056911 | 0.001248 | 0.005749 | 0.003356 | 0.999202 | 0.023893 | 0.023893 | $5.708795 \times 10^{-4}$ | 0.023893 | $5.103174 \times 10^{-11}$ | $1.892869 \times 10^{-14}$ | $1.364007 \times 10^{-5}$ | 0.003060 | $1.943214 \times 10^{-6}$ |
| 8 | 0.078352 | 0.122611 | 0.000000 | 0.078389 | 0.155148 | 0.097674 | 0.999810 | 0.086568 | 0.086568 | $7.494039 \times 10^{-3}$ | 0.086568 | $1.072959 \times 10^{-12}$ | $1.098443 \times 10^{-16}$ | $6.487448 \times 10^{-4}$ | 0.000437 | $3.645383 \times 10^{-6}$ |
| 9 | 0.208805 | 0.000000 | 0.000000 | 0.000000 | 0.000000 | 0.000000 | 0.000000 | 0.151202 | 0.151202 | $2.286197 \times 10^{-2}$ | 0.151202 | $2.013662 \times 10^{-13}$ | $1.180272 \times 10^{-17}$ | $3.456769 \times 10^{-3}$ | 0.093885 | $2.387736 \times 10^{-3}$ |
| 10 | 0.208805 | 0.289809 | 0.024390 | 0.176494 | 0.395984 | 0.282752 | 0.999915 | 0.196320 | 0.196320 | $3.854167 \times 10^{-2}$ | 0.196320 | $9.199435 \times 10^{-14}$ | $4.152869 \times 10^{-18}$ | $7.566513 \times 10^{-3}$ | 0.043935 | $1.883721 \times 10^{-3}$ |
| 11 | 0.176494 | 0.648487 | 0.024390 | 0.045932 | 0.180633 | 0.115217 | 0.999711 | 0.091531 | 0.091531 | $8.377937 \times 10^{-3}$ | 0.091531 | $9.077188 \times 10^{-13}$ | $8.788911 \times 10^{-17}$ | $7.668416 \times 10^{-4}$ | 0.010724 | $9.994741 \times 10^{-5}$ |
| 12 | 0.034880 | 0.000000 | 0.000000 | 0.000000 | 0.000000 | 0.000000 | 0.000000 | 0.000000 | 0.000000 | 0.000000 | 0.000000 | 0.000000 | 0.000000 | 0.000000 | 0.000000 | 0.000000 |
| 13 | 0.009289 | 0.000000 | 0.000000 | 0.000000 | 0.000000 | 0.000000 | 0.000000 | 0.006727 | 0.006727 | $4.524683 \times 10^{-5}$ | 0.006727 | $2.287045 \times 10^{-9}$ | $3.013234 \times 10^{-12}$ | $3.043563 \times 10^{-7}$ | 0.002244 | $1.127246 \times 10^{-7}$ |
| 14 | 0.017367 | 0.000000 | 0.000000 | 0.017403 | 0.938259 | 0.905281 | 0.999816 | 0.016768 | 0.016768 | $2.811557 \times 10^{-4}$ | 0.016768 | $1.476511 \times 10^{-10}$ | $7.803968 \times 10^{-14}$ | $4.714334 \times 10^{-6}$ | 0.004492 | $1.404966 \times 10^{-6}$ |
| 15 | 0.000000 | 0.000000 | 0.000000 | 0.000000 | 0.000000 | 0.000000 | 0.000000 | 0.000000 | 0.000000 | 0.000000 | 0.000000 | 0.000000 | 0.000000 | 0.000000 | 0.000000 | 0.000000 |

*3.4. Model Training*

Once statistical features were extracted and data normalized, learning models for predicting non-active states of the machines were constructed by combining the raw data from the sensor device and extracted features. Considering that we deal with a dataset that contains active and non-active states, we can formulate the predicting non-active states of machines as a classification problem. Thus, we constructed predictive models using the following state-of-the-art machine learning methods suitable for classification: Decision tree, kNN, random forest, and linear SVM.

A decision tree is one of the most popular methods used for classification. The decision tree predicts the class of a target feature by producing simple decision rules inferred from the data [32]. In the context of the proposed method, these decision rules enabled us to understand and interpret non-active states of machines easily. kNN classification was one of the most fundamental and simple classification methods. It classifies the class of a target feature by using the distance between a test sample and the specified training samples [33]. kNN is suitable for datasets with numeric values. Thus, it can be easily adapted for our dataset as it contains numeric values obtained through statistical analysis. Breiman [34] introduced the random forest method, which was a combination of tree predictors. In other words, a random forest algorithm creates a decision tree on data samples, gets the prediction from each of them, and finally selects the best solution. Despite its complexity, the random forest algorithm achieves high accuracy because it creates multiple trees and enables us to choose a tree with the highest accuracy. Considering that predicting the non-active status of machines with high accuracy can bring benefits to the efficient production of manufacturing machines, we can efficiently utilize the random forest algorithm to solve our problem. SVM was another classification method that was widely used for the analysis of a large amount of data. One of the main advantages of SVM was that it was effective in high dimensional spaces. In the case of our dataset, we obtained 17 input features, including raw data from the sensor device. Considering that the proposed dataset was relatively high dimensional, we can benefit from using SVM for classifying the non-active state of the machines. Table 5 demonstrate the advantages and disadvantage of classification algorithm used in this study.

**Table 5.** The pros and cons of classification methods.

| Method | Pros | Cons |
|---|---|---|
| Decision Tree | • Simple to understand and to interpret<br>• Requires little data preparation<br>• Handle numerical and categorical data | • Easy overfitting.<br>• The greedy algorithm<br>• Can be unstable |
| KNN | • Simple to understand and to interpret<br>• Useful for non-linear data<br>• A versatile algorithm | • A bit expensive algorithm<br>• High memory storage required.<br>• Very sensitive |
| Random Forest | • Overcomes the problem of overfitting<br>• Very flexible<br>• Less variance than single decision tree | • Complexity<br>• More computational resources are required |
| Linear SVM | • Effective in high dimensional spaces.<br>• Memory efficient<br>• Versatile | • Easy to overfitting<br>• Do not directly provide probability estimates |

## 4. Performance Evaluation

*4.1. Experimental Setup*

The methods and techniques to select the appropriate operating pattern subject to the optimal value of power per product quantity are described herein. In addition, to obtain the experimental results, we used a computer with a CPU Intel® Core™ i7-6700 3.40 GHz, 32 GB of RAM, NVIDIA GeForce 9800 GT graphics card, Windows 10 operating system, and an integrated development

environment Jupyter notebook for Python and IntelliJ for Java. Moreover, the Maria database was used to store a large amount of data.

### 4.2. Hyperparameter of Competing Methods

Recall from Section 3.4 that we constructed predictive models using a decision tree, kNN, random forest, and linear SVM. Table 6 demonstrates a detailed description of each hyperparameter of competing methods. Various combinations of the parameters were applied in the experiments, and the most optimum set was selected. First, the prediction of the non-active state of machines with a decision tree was performed using the Gini impurity. Further, the kNN model was constructed using the Minkowski distance to compute the distance between an observation data point and its nearest neighbors with $k = 5$. Here, we tested various combinations of $k$ and selected as $k = 5$ because it produced the highest accuracy. Moreover, utilizing the advantage of the bagging method, the random forest was developed using the Gini impurity. Finally, the radial basis function, a kernel SVM, was employed to map the data into a high-dimensional space. Here, the C parameter, which controls the penalty of the data points outside the margin, was set to 10. Similar to the k parameter of kNN, we tested various combinations of C parameter and selected as C = 10 as it produced the highest accuracy.

**Table 6.** The hyperparameters of competing methods.

| Algorithm | Parameters | |
|---|---|---|
| | class_weight | None |
| | criterion | gini |
| | max_depth | None |
| | max_features | None |
| | max_leaf_nodes | None |
| | min_impurity_decrease | 0.0 |
| **Decision Tree** | min_impurity_split | None |
| | min_samples_leaf | 1 |
| | min_samples_split | 2 |
| | min_weight_fraction_leaf | 0.0 |
| | presort | False |
| | random_state | None |
| | splitter | best |
| | algorithm | auto |
| | leaf_size | 30 |
| | metric | minkowski |
| | metric_params | None |
| **KNN** | n_jobs | None |
| | n_neighbors | 5 |
| | p | 2 |
| | weights | uniform |
| | class_weight | None |
| | criterion | gini |
| | max_depth | None |
| | max_features | None |
| | max_leaf_nodes | None |
| | min_impurity_decrease | 0.0 |
| **Random Forests** | min_impurity_split | None |
| | min_samples_leaf | 1 |
| | min_samples_split | 2 |
| | min_weight_fraction_leaf | 0.0 |
| | presort | False |
| | random_state | None |
| | splitter | best |

**Table 6.** *Cont.*

| Algorithm | Parameters | |
|---|---|---|
| | C | 10 |
| | class_weight | None |
| | dual | True |
| | fit_intercept | True |
| | intercept_scaling | 1 |
| **Linear SVM** | loss | squared_hinge |
| | max_iter | 1000 |
| | multi_class | ovr |
| | penalty | l2 |
| | random_state | None |
| | tol | 0.0001 |
| | verbose | 0 |

### 4.3. Evaluation Metrics

The four machine learning methods were validated based on the root mean square error (RMSE), precision and recall scores, and F1 scores.

We first describe the *RMSE*. The *RMSE* computes the variance between values predicted by a hypothetical model and the real values. In other words, it measures the standard-of-fit between the actual data and the predicted model. It is expressed as follows:

$$RMSE = \sqrt{\frac{\sum_{i=1}^{n}(\hat{y}_i - y_i)^2}{n}}, \tag{9}$$

In Equation (9), $\hat{y}_i$ is the predicted variable, $y_i$ the actual variable, and $n$ the number of observations. The *RMSE* is always non-negative. In general, a lower *RMSE* is better.

Precision, recall, and F1 score are measurements to check the accuracy of training models. Specifically, precision is a measure of result relevancy, whereas recall represents the degree to which many genuinely relevant results are returned. Meanwhile, the F1 score is calculated using the values obtained from precision and recall. Precision, recall, and F1 score are expressed as follows:

$$Precision = \frac{\sum_{i=1}^{n} TP/(TP+FP)}{n}$$

$$Recall = \frac{\sum_{i=1}^{n} TP/(TP+FN)}{n} \tag{10}$$

$$F_1 \ score = \frac{TP}{TP + \frac{1}{2}(FP+FN)}$$

In Equation (10), *TP*, *FP*, and *FN* indicate true positive, false positive, and false negative, respectively; $n$ represents the total amount of observation data.

### 4.4. Experimental Results

Table 7 presents the experimental results. Here, for the training dataset, we used 86,400 samples, where 73,019 active states and 13,381 non-active states were detected.

**Table 7.** Model Comparison Result.

| Model | Precision | Recall | F1 Score | Observations | RMSE |
|---|---|---|---|---|---|
| Decision tree | 92% | 75% | 83% | 86,400 | 0.3478 |
| KNN | 93% | 95% | 94% | 86,400 | 0.2085 |
| Random forest | 96% | 85% | 90% | 86,400 | 0.1813 |
| Linear SVM | 98% | 99% | 98% | 86,400 | 0.0677 |

From the table, we can observe that linear SVM achieved 98% accuracy in terms of F1 score in the classification of non-active states of machines, followed by kNN, random forest, and decision tree, which were only 94%, 90%, and 83%, respectively. We can also observe from the table that linear SVM also achieved the lowest error rate (i.e., 0.0677) compared with other methods. Linear SVM used two hyperplanes that separated the observations linearly, thus that there were no observations between them. Linear SVM performed better than other methods in our experiments as the dataset has been constructed using features that are suitable for creating linearly-separable observations. On the other hand, the random forest also performed well, which resulted in accuracy close to SVM in terms of precision. Random forest is a well-known knowledge-based ensemble method that overcomes the overfitting problem as well as errors due to bias in the decision tree and, therefore, yields high accuracy. As kNN is not a knowledge-based ensemble technique like the random forest, it achieves marginally less accurate results but still outperforms the decision tree in terms of precision, recall, and F1 score.

Figure 8 demonstrates the visualized result of SVM. In Figure 9a,b, we can see machine states for a single day. Here, the blue line indicates the actual machine state (in every second), and the red line indicates predicted machine states (in every second) using linear SVM. From the figures, it is clear that the prediction by linear SVM follows the actual machine state well. On the other hand, the result also suggests that there were some errors (e.g., approximately at 02:00, 04:00, or 05:30) in the prediction process. However, a closer look for these errors indicated that wrongly predicted observations were short-term (a matter of seconds) and, therefore, were insignificant to the overall prediction of non-active state predictions.

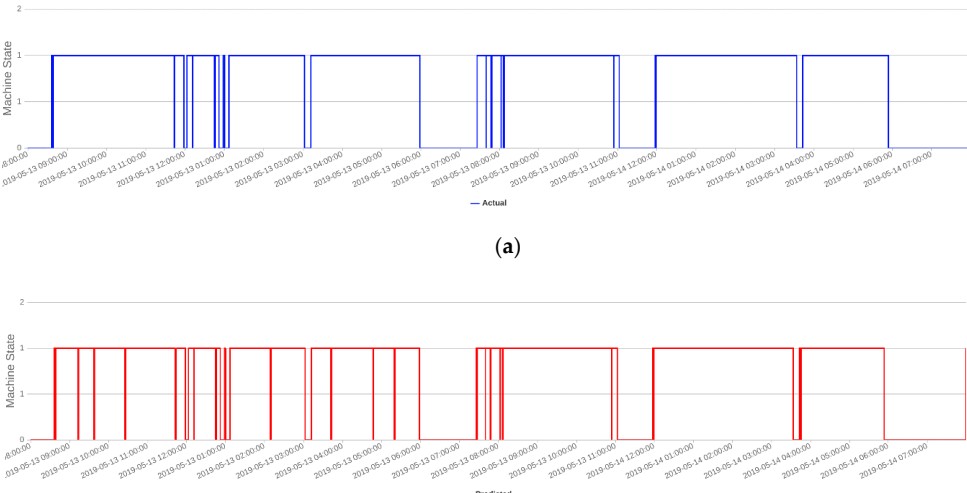

**Figure 9.** Comparison of the actual and predicted results of the support vector machine (SVM). (**a**) Actual machine states (in every second). (**b**) Predicted machine states (in every second) using linear SVM.

## 5. Conclusions

In this paper, a combination of various statistical methods and machine learning have been proposed to predict the active and non-active states of the machine. In this approach, reliability analysis, the proposed feature extraction method MSTV, as well as time- and frequency-domain analyses, were used to extract features from a raw dataset. Hence, 17 useful features were obtained in the extraction phase, including Weibull, lognormal, and exponential probability distributions, and the proposed feature extraction the MV, RMSV, SMRV, KC, SF, SC, which will be utilized to extract the new features from the MSTV. In addition, RMSF and RVF. Furthermore, after normalization, we used to train four types of machine learning, i.e., the decision tree, kNN, random forest, and a linear SVM model to detect the active and non-active states of the machine. According to the experimental results, the linear SVM model's predicted result indicated a 98% accuracy rate compared with the actual value;

this demonstrated that the model was highly accurate and, therefore, can be utilized to reduce machine faults and increase productivity.

Even though the performance of the non-active state detection model was satisfactory with high accuracy, the statistical method to extract the train features was conservative, and the detection model could not solve all the issues encountered in manufacturing. Therefore, in the future, we plan to utilize more statistical methods such as the Hilbert–Huang Transform or the fast Fourier transform along with the proposed approach to extract more valuable features and develop a promising non-active state prediction model. Finally, we will attempt to create a user-friendly interface for displaying the results in a web application platform.

**Author Contributions:** Conceptualization, K.-H.Y.; methodology, T.B.; software, A.N.; validation, formal analysis, T.B.; investigation, T.B.; resources, T.B.; data curation, T.B.; writing—original draft preparation, T.B.; writing—review and editing, T.B.; supervision, S.B.; software-upgrade and data generation, K.-H.Y.; project administration, K.-H.Y.; funding acquisition, K.-H.Y. All authors have read and agreed to the published version of the manuscript.

**Funding:** This research was supported by the Technology Innovation Program (2004367, development of cloud big data platform for the innovative manufacturing in ceramic industry) funded by the Ministry of Trade, Industry & Energy (MOTIE, Korea) and by the MSIT (Ministry of Science and ICT), Korea, under the Grand Information Technology Research Center support program (IITP-2020-0-01462) supervised by the IITP (Institute for Information & communications Technology Planning & Evaluation)

**Acknowledgments:** We also appreciate the efforts of Tserenpurev Chuluunsaikhan, a PhD student in Department of Computer Science at Chungbuk National University, who helped in revision of this paper.

**Conflicts of Interest:** The authors declare no conflict of interest.

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
