# Peer review of "Prediction of Machine Inactivation Status Using Statistical Feature Extraction and Machine Learning"

_applsci, doi:10.3390/app10217413_

Round 1

Reviewer 1 Report

Please consider to shorten the introduction, moving some sentences about the state of the art to the section 2. Indeed, the specific aim of the paper should be better declared, it may be not sufficiently clear.

Please, add explanations of the colors in Figure 1.

The sentences at the lines 325-330 are a repetitions of already said things.

At line 334 explain as weights have been determined.

Define KNN acronym.

The results should be discussed more extensively.

Author Response

Dear Reviewer,

Thank you for your kind review.

We attached responses of your comments.

Thank you very much.

Kwan-Hee Yoo

Reviewer 2 Report

This is a well written paper. Just few points:

  1. K = 5 was chosen for the KNN model. Would you please state why.
  2. C= 10 was selected for SVM - again, please elaborate how you or why you selected C=10.
  3. The numbering system should have the comma after every three numbers, so the numbers should be:86,400 and not 8,6400

Author Response

(The authors gave the same response as above.)

Reviewer 3 Report

This paper presents a comparative study using 4 different ML techniques for machine inactivation status prediction. The contribution of this paper is more towards the applicable side rather than algorithm novelty. Before this paper can be considered for publication, I have several comments for the authors.

  1. Firstly, the introduction need to be modified significantly due to the lack of flow and duplicated description.
    • Line 30 - The authors mentioned that continuous flow processes can reduce inventory and transportation cost and increase productivity. Followed by mentioning In addition, massive industrial processes can be detrimental to machines. How does the second sentence constitute to a "in addition" when they have no link.
    • Line 34 to 36 - Authors need to provide references for the models or methods suggested in the literature.
    • Line 39 - Authors should replace despite with Beside
    • Line 40 - Authors mentioned that it is difficult to construct a quality detection model but fail to mention why
    • Line 53 - Authors should provide reference for the python based framework 
    • Line 58 onwards seems to be better written but is repeating what was mentioned before in different words. I have no idea why is this the case. 
  2. Section 2
    • Line 116 and 126 - authors not author and provide reference immediately
    • Line 118 which describe the weibull distribution all the way to equation 2 is redundant. Since this is related studies, authors should summarized what was done and provide equations only when necessary. 
    • Line 129 - 131 - Authors copied the symbols from the reference image and fail to rephrase it according to their context and we have a lot of symbols that did not appear in text.
    • Equation 3 is wrong
    • Line 134, the reference provide their accuracy in r-square, there shouldn't be a percentage behind, otherwise it is ridiculous to have a model where the accuracy is only 0.866%.
    • Line 153 - 22 features were extracted in the paper, not 21
    • Line 153 - its z-score higher than 0.6 not score higher than 0.6
    • Not sure what Figure 2.4 (Line 155) and Figure 2.5 (161) and Figure 2.6 (line 173) are referring to as they are not in text 
    • Line 162 - I suggest rewriting to this "TTF prognostics began with low accuracy and will only become higher with more data"
    • Line 186 - I strongly disagree with the statement "approaches proposed [8,12] are extremely simple and consider only a few factors to build a detection model, resulting in low accuracy". Simple model do not necessary means it is lousy. Please rephrase this accordingly.
  3.  Section 3
    • Line 199 - what do the authors mean by “use an alarm that notifies the machine of an abnormality while the machine is operating”
    • Line 208 – what table are constructed ? and this paragraph should be at the start of the section
  4. Figure 13 – Text should be centered for proper presentation
  5. Entire section 6 is redundant. Authors only need to mention they are using which ML techniques and what hyperparameters they are using.
  6. Section 7
    •  there is no discussion on the result except stating the best model is SVM. There should be some form of analysis provided. For example, the time to compute the result and etc.
    • Since precision and recall are given, why not F1-score? Since F1-score is a better or more universal metric
    • Figure 18 actually doesn’t show much. Authors can consider including some more pictures of the GUI to indicate what will happen when non-active is detected
  7.  References are not in proper format and are inconsistent. Moreover, there are some references that are cited from unreliable sources such as Wikipedia.

Author Response

(The authors gave the same response as above.)

Round 2

Reviewer 3 Report

The paper has been properly revised accordingly to my comments and I do not have any further comments.